# Transcriptome and Differentially Expressed Gene Profiles in Mycelium, Primordium and Fruiting Body Development in *Stropharia rugosoa**n**nulata*

**DOI:** 10.3390/genes13061080

**Published:** 2022-06-17

**Authors:** Haibo Hao, Jinjing Zhang, Qian Wang, Jianchun Huang, Jiaxiang Juan, Benke Kuai, Zhiyong Feng, Hui Chen

**Affiliations:** 1National Research Center for Edible Fungi Biotechnology and Engineering, Key Laboratory of Applied Mycological Resources and Utilization, Ministry of Agriculture, Shanghai Key Laboratory of Agricultural Genetics and Breeding, Institute of Edible Fungi, Shanghai Academy of Agricultural Sciences, Shanghai 201403, China; hhb199232@163.com (H.H.); wq-15309@163.com (Q.W.); jianmushroom@163.com (J.H.); jiaxiang_j2012@163.com (J.J.); feng_zy@yahoo.com (Z.F.); 2State Key Laboratory of Genetic Engineering and Fudan Center for Genetic Diversity and Designing Agriculture, Institute of Plant Biology, School of Life Sciences, Fudan University, Shanghai 200438, China; bkkuai@fudan.edu.cn; 3College of Life Science, Nanjing Agricultural University, Nanjing 210095, China

**Keywords:** *Stropharia rugosoannulata*, differentially expressed genes, carbohydrate enzyme genes, environmental factors, transcription factors, fruiting body development

## Abstract

*Stropharia rugosoannulata* uses straw as a growth substrate during artificial cultivation and has been widely promoted in China. However, its fruiting body formation and development processes have not been elucidated. In this study, the developmental transcriptomes were analyzed at three stages: the mycelium (G-S), primordium (P-S) and fruiting body (M-F) stages. A total of 9690 differentially expressed genes (DEGs) were identified in the different developmental stages. Gene Ontology (GO) and Kyoto Encyclopedia of Genes and Genomes (KEGG) enrichment analyses showed that these DEGs were involved mainly in hydrolase activity, structural molecule activity and oxidoreductase activity as well as xenobiotic biodegradation and metabolism and energy metabolism pathways. We further found that the higher expression of most carbohydrate enzyme (i.e., GH, CE, CBM, AA and PL) genes in the hyphal (i.e., G-S) stage was related mainly to substrate degradation, while the upregulation of glycosyltransferase (GT) gene expression in the P-S and M-F stages may be related to cell wall synthesis. In addition, we found that CO_2_-sensing-related genes (i.e., *CA-2*, *CA-3*, *PKA-1* and *PKA-2*) were upregulated in the P-S and M-F stages, heat shock protein genes (*HSP60* and *HSP90*) were significantly downregulated in the P-S stage and upregulated in the M-F stage and the transcription factors (i.e., *steA*, *MYB*, *nosA*, *HAP1*, and *GATA-4/5/6*) involved in growth and development were significantly upregulated in the P-S stage. These results suggest that environmental factors (i.e., CO_2_ and temperature) and transcription factors may play a key role in primordium formation. In short, this study provides new insights into the study of stimulating primordia formation affecting the development of fruiting bodies of *S. rugosoannulata.*

## 1. Introduction

*S*. *rugosoannulata* Farl. ex Murrill, commonly known as the wine-cap *Stropharia* mushroom or king *Stropharia*, is a nutritional mushroom that is widely distributed in northern temperate zones [1,2]. *S. rugosoannulata* is recommended by the Food and Agriculture and Organization (FAO) for cultivation in developing countries [3]. *S. rugosoannulata* can be grown on various raw materials, such as rice straw, wheat straw, corn straw and dead tree branches, and the mushroom is cultivated in fields, woodlands and simple greenhouses. In recent years, studies have shown that the cultivation of *S. rugosoannulata* can increase soil organic carbon by 57.4–61.6% in topsoil (0–10 cm) and by 24.8–39.9% in subsoil (10–30 cm) [4]. However, the molecular mechanisms underlying the formation and development of fruiting bodies remain unclear.

To study the growth and development process of *S. rugosoannulata*, we optimized a set of appropriate operating procedures from mycelial growth to fruiting body maturation under factory conditions including light, temperature, humidity and carbon dioxide (CO_2_) concentration control. Compared with outdoor cultivation methods, our factory standard operation method will be better for studying the life history of *S. rugosoannulata.* The life cycle of *Agaricus bisporus* includes two stages, vegetative hyphae and fruiting body formation, and the vegetative hyphae are considered to provide nutrients for the growth of fruiting bodies [5]. *S. rugosoannulata* is similar to *A. bisporus*, and both fungi use straw as a substrate and share many similarities in their life histories. The carbohydrate active enzyme (CAZyme) that degrades the substrate has been widely studied, and the CAZyme of *A. bisporus* during the vegetative hyphal stage and the fruiting body stage has also been analyzed in detail [5]. Moreover, CAZyme genes related to the degradation of plant biomass components are mostly involved in the hyphal stage, while fruiting bodies mainly express CAZyme genes related to the synthesis and modification of fungal cell wall [5]. In addition, a comparison of the CAZyme genes of the three litter/straw-degrading species (i.e., *A. bisporus*, *Volvariella volvacea* and *Coprinopsis cinerea*) revealed that they have similar compositions, but there were significant differences in the number of genes involved [6]. Therefore, it is necessary to further study whether the composition and number of CAZyme genes of *S. rugosoannulata* are different from those of other litter/straw-degrading fungi.

Studies have shown that a suitable temperature for the growth of vegetative hyphae of *S. rugosoannulata* is 5–32 °C, the primordium does not differentiate at temperatures over 25 °C and the fruiting body quality deteriorates at temperatures over 20 °C in outdoor cultivation [7]. Mushroom-forming fungi differentiate by sensing several environmental factors (i.e., nutrient, temperature, CO_2_ concentration and light conditions) for fruiting body formation [8]. Starvation is a critical signal of environmental deterioration. Therefore, nutrients are critical signals for sexual reproduction in mushroom-forming fungi [8]. Studies have shown that fruiting bodies are more sensitive to CO_2_ concentrations in the early stages of development and that high CO_2_ affects the synthesis of the cell wall component R-glucan and fruiting body cell morphology [9,10,11]. Moreover, temperature decrease-induced fruiting body formation has been applied in many mushrooms [12] such as *Lentinula edodes* [13], *A. bisporus* [14] and *Flammulina velutipes* [15]. Light can also induce fruiting bodies or promote fruiting body production, such as in *L. edodes* [16], *Polyporus* (*Favolus*) *arcularius* [17] and *C. cinerea* [18]. However, the presence of light may not always be essential for the induction of fruiting bodies [8]. Interestingly, previous research has shown that the formation of fruiting bodies in *S. rugosoannulata* is not sensitive to light [19].

At present, RNA-seq analysis has been widely used in fruiting body development research on various edible and medicinal fungi including *S. commune* [20], *A. bisporus* [14], *Ganoderma lucidum* [21], *C. cinerea* [22], *F. velutipes* [23], *V. volvacea* [24], *L. edodes* [25] and *Chinese cordyceps* [26]. However, unlike in ascomycetes, the molecular mechanism for fruiting body development in basidiomycetes is not well understood [8], mainly due to the long cultivation cycle of basidiomycetes and the participation of various environmental factors in the formation and development of fruiting bodies.

In this study, nine samples of *S. rugosoannulata* at different growth development stages were used as materials for RNA-seq analysis. Based on a combination of GO, KEGG and qRT–PCR correlation analyses, this study identified differentially expressed genes (DEGs) in the growth development stages. In particular, we focused on the differential expression of CAZyme genes, CO_2_-responsive genes, heat shock proteins and transcription factors. This work provides important insights into *S. rugosoannulata* development and the environmental factors that affect the formation and development of fruiting bodies. It also provides a theoretical foundation for the factory production of *S. rugosoannulata.*

## 2. Materials and Methods

### 2.1. Collection of S. rugosoannulata Materials at Different Developmental Stages

The *S. rugosoannulata* strain, DQ-1 (CGMCC5.2211), was deposited in the China General Microbiological Culture Collection Center. The solid medium consisted of 50% corncob, 30% sawdust, 15% rice bran, 4% wheat bran and 1% calcium carbonate. After complete mixing, the substrate was packed into polypropylene cultivation bags (average of 1000 g/bag with a moisture content of 63–65%), sterilized at 121 °C for 2.5 h and inoculated with a pure culture of *S. rugosoannulata.* Then, the cultivation bags were kept at 22–25 °C and 65–70% relative humidity in the dark until the cultivation substrate was fully covered by mycelia. Then, 2.5 kg of rice straw (moisture content 68%) was transferred to each cultivation basket, and pure culture mycelium of *S. rugosoannulata* was inoculated. After cultivation in the dark for 25 days, the cultivation basket was placed in the factory workshop at Shanghai Guosen Biotech Co., Ltd. (Shanghai, China), for cultivation. Temperature, humidity and CO_2_ concentration were controlled and recorded by the intelligent IoT system provided by Agricultural Engineering Manufacturing (AEM, Maasbree, The Netherlands), and the control parameters were corrected on the basis of measurements made with a thermometer, hygrometer and CO_2_ detector. Samples were collected at the mycelial stage (25 d), primordium stage (35 d) and fruiting body stage (45 d) of cultivation and frozen at −80 °C. Three biological replicates of each stage were used for the RNA-seq analysis and subsequent experiments.

### 2.2. Total RNA Isolation, cDNA Library Preparation and Illumina Sequencing

Total RNA was extracted from the samples using TRIzol^®^ Reagent according to the manufacturer’s instructions (Invitrogen, Carlsbad, CA, USA), and genomic DNA was removed using DNase I (TaKaRa, Dalian, China). Then, RNA quality was determined using a 2100 Bioanalyzer (Agilent, Palo Alto, CA, USA) and quantified using an ND-2000 system (NanoDrop Technologies, Wilmington, DE, USA). A high-quality RNA sample (OD260/280 = 1.8~2.2, OD260/230 ≥ 2.0, RIN ≥ 6.5, 28 S/18 S ≥ 1.0, >10 μg) was used to construct a sequencing library. RNA-seq transcriptome libraries were prepared using a TruSeq^TM^ RNA Sample Preparation Kit from Illumina (San Diego, CA, USA) with 1 μg of total RNA. Briefly, messenger RNA was isolated with poly(A) selection by oligo(dT) beads and fragmented using fragmentation buffer. cDNA synthesis, end repair, A-base addition and ligation of the Illumina-indexed adaptors were performed according to Illumina’s protocol. Libraries were then size-selected for cDNA target fragments of 200–300 bp on 2% Low Range Ultra Agarose followed by PCR amplification using Phusion DNA polymerase (NEB) with 15 PCR cycles. After quantification by TBS380, paired-end libraries were sequenced by Illumina NovaSeq 6000 sequencing (150 bp*2, Shanghai BIOZERON Co., Ltd, Shanghai, China). The datasets presented in this study can be found online in the NCBI repository (https://www.ncbi.nlm.nih.gov/ (accessed on 19 March 2021)) under accession number: SRP311165.

### 2.3. Read Quality Control and Mapping

The raw paired-end reads were trimmed and quality controlled using Trimmomatic with the following parameters: SLIDINGWINDOW, 4, and 15 MINLEN, 75 (version 0.36) (http://www.usadellab.org/cms/uploads/supplementary/Trimmomatic (accessed on 1 June 2020)). Then, clean reads were separately aligned to the reference genome in the orientation mode using HISAT2 software (https://ccb.jhu.edu/software/hisat2/index.shtml (accessed on 1 June 2020)). This software was used to map the default parameters. Quality assessment of these data was performed with Qualimap v2.2.1 (http://qualimap.bioinfo.cipf.es/ (accessed on 3 June 2020)), and HTSeq (https://htseq.Readthedocs.io/en/release_0.11.1 (accessed on 3 June 2020)) was used to count the reads of each gene.

### 2.4. Differential Expression Analysis and Functional Enrichment

To identify DEGs between the two different growth stages of *S. rugosoannulata*, the expression level of each gene was calculated using the fragments per kilobase of exon per million mapped reads (FRKM) method. The R statistical package “edgeR” (Empirical analysis of Digital Gene Expression in R) (http://www.bioconductor.org/packages/release/bioc/html/edgeR.html/ (accessed on 5 June 2020)) was used for the differential expression analysis. The DEGs between two samples (P-S vs. G-S, M-F vs. P-S and M-F vs. G-S) were selected using the following criteria: a logarithmic fold change greater than 2 and a false discovery rate (FDR) less than 0.05. To understand the functions of the DEGs, GO functional enrichment and KEGG pathway analysis were carried using Goatools (https://github.com/tanghaibao/Goatools (accessed on 6 June 2020)) and KOBAS (http://kobas.cbi.pku.edu.cn/home.do (accessed on 6 June 2020)). DEGs were significantly enriched in GO terms and metabolic pathways when their Bonferroni-corrected *p*-value was less than 0.05.

### 2.5. Validation of Gene Expression by Quantitative Real-Time Polymerase Chain Reaction (qRT–PCR)

Approximately 2 μg of total RNA from *S. rugosoannulata* at the three growth stages (i.e., the hyphal (G-S), primordium (P-S), and mature-fruiting (M-F) stages) was reverse-transcribed by M-MLV reverse transcriptase (Takara, Dalian, China) using oligo (dT) as the primer. qRT–PCR was performed using SYBR (Takara, Dalian, China) [27]. The primers and internal reference gene (18 S ribosomal RNA) are listed in Appendix A. Moreover, relative gene expression was analyzed using the 2^−ΔΔCt^ method [28], and each experiment was performed in triplicate.

### 2.6. Statistical Analysis

All experimental data presented in this paper are based on three independent samples to ensure that the trends and relationships observed in the cultures were reproducible. The data and graphs were processed using GraphPad Prism 6.0. Differences among treatments were analyzed by one-way analysis of variance (ANOVA) combined with Duncan’s multiple range test at a probability of *p* < 0.05.

## 3. Results

### 3.1. Analysis of the Morphological Features of S. rugosoannulata

Under factory cultivation conditions, the mycelium completely covered the cultivation substrate after culturing at 23 °C for 25 days (Figure 1A). To stimulate the formation of primordia, the temperature was gradually lowered from 23 to 12 °C, the carbon dioxide concentration was also gradually reduced (3000 to 2000 ppm), and the humidity was increased (60% to 95%). When the mycelium climbed to the surface of the peat soil, the temperature was gradually increased (12 to 15 °C), which promoted mycelial kinking and the formation of a spherical primordium with a white villus appearance and hard texture (Figure 1). After the primordium formed, the temperature was kept at 15 °C, the carbon dioxide was 2000 ppm, and the air humidity was greater than 80% (Figure 1B–D). The mushroom cap gradually turned from white to dark red, the annulus formed and extended outward, and tiny scale formations were designated mature fruiting bodies (Figure 1A).

### 3.2. Global Transcriptomic Analysis of S. rugosoannulata and Identification of DEGs

To describe the patterns of gene expression during growth and development, nine libraries were constructed using samples from three different growth stages of *S. rugosoannulata*. A total of 558.99 million raw reads were generated by Illumina sequencing. After applying cleaning and quality control steps, 528.31 million clean reads were obtained, and the Q30 value of the base ratio was higher than 94.16% (Appendix A). Moreover, 91.65–94.75% of the reads could be mapped to the *S. rugosoannulata* genome (Appendix A). The obtained RNA sequences were assembled using the sequence clustering software Trinity, with 11,459 transcripts and 9646 genes annotated via the NR, STRING gene, GO, COG, KEGG, and SWSS databases with an E-value of 10^−5^. A total of 83.9% were annotated as known functional genes.

Based on a DEG analysis of the transcriptomes of the consecutive developmental stages of *S. rugosoannulata*, a total of 2969 DEGs were identified between the P-S and G-S stages, which covered 25.82% of the annotated genes, with 1679 genes being upregulated and 1290 being downregulated (Figure 2A). A total of 3287 DEGs were identified between the M-F and P-S stages, and the numbers of upregulated genes (1660) and downregulated genes (1627) were not very different (Figure 2A). The largest number of DEGs was identified between the M-F and G-S stages (3434), and the number of upregulated genes (1907) was significantly greater than the number of downregulated genes (1527). A Venn diagram was used to analyze the DEGs between stages, and 709 DEGs were found to overlap among the three groups. Moreover, the M-F and P-S comparisons had the largest number of unique DEGs with 774 genes (Figure 2B). Further analysis of the expression patterns of the 709 overlapping DEGs showed that some DEGs were significantly downregulated after the G-S stage, but more genes were upregulated in the P-S and M-F stages, and the gene expression patterns in the P-S and M-F stages were more similar (Figure 2C). The overall gene expression patterns suggested differences in gene expression among the growth stages, but the P-S and M-F stages were the most similar (Figure 2D). These results revealed that a large number of DEGs are required to complete the process from mycelial kinking to fruiting bodies and further development.

### 3.3. GO Enrichment and KEGG Pathway Analysis for DEGs

Based on GO functional classification, all DEGs were classified into three different categories: biological process (BP), cellular component (CC) and molecular function (MF). Then, with a *p*-value ≤ 0.05 as the threshold for significant enrichment, the classifications with significant differences were identified. The top 30 categories for each comparison are displayed in Appendix A, and the results show that in the comparison of P-S vs. G-S, the significantly enriched terms involved in MFs and BPs included hydrolase activity (543), small molecule metabolic process (391), small molecule biosynthetic process (198) and carbohydrate derivative metabolic process (174) (Figure 3). This finding indicates that the substrate needs to be degraded to provide nutrients for growth in the process of mycelium kinking to form fruiting bodies. In the M-F vs. P-S stage comparison, the processes that involved more genes included structural molecule activity (165), organelle fission (135), ribosome (119), mitochondrial protein complex (102), structural constituent of ribosome (100) and meiosis I (67) (Figure 3), indicating that a large amount of energy synthesis is involved in the growth and development of fruiting bodies to promote cell proliferation, differentiation and sexual reproduction. In addition, the M-F vs. G-S comparisons were associated mainly with oxidoreductase activity (275), carbohydrate metabolic processes (169), chromosome segregation (129) and extracellular regions (112) (Figure 3), also indicating the degradation of the substrate, the meiotic process of fruiting body sexual reproduction and the oxidative stress process of cells being challenged by environmental factors occurred from the G-S stage to the M-F stage.

The DEGs were mapped to the KEGG database, and an enrichment analysis was performed to determine their functions. The top 20 KEGG pathways with significant enrichment of these DEGs are shown in Appendix A. The P-S vs. G-S comparison involved mainly phenylpropanoid biosynthesis (ko00940), cyanoamino acid metabolism (ko00460), metabolism of xenobiotics by cytochrome P450 (ko00980), polycyclic aromatic hydrocarbon degradation (ko00624) and naphthalene degradation (ko00626). These pathways belong mainly to xenobiotic biodegradation and metabolism and the biosynthesis of other secondary metabolites, which also suggests that the mycelial stage involves mainly the degradation of culture substrates and synthesis of secondary metabolites. The ribosome pathway (ko03010), an important pathway for protein synthesis, was significantly enriched in the M-F vs. P-S comparison. Moreover, the oxidative phosphorylation (ko00190), proteasome (ko03050), biosynthesis of unsaturated fatty acids (ko01040) and meiosis–yeast (ko04113) pathways were also significantly enriched during the development of fruiting bodies. In addition, in the M-F and G-S comparisons, we found that the DNA replication (ko03030), mismatch repair (ko03430), glycosphingolipid biosynthesis (ko00603), meiosis-yeast (ko04113), and amino sugar and nucleotide sugar metabolism (ko00520) pathways were mainly enriched (Appendix A). Overall, the growth process from vegetative mycelium to mature fruiting body involved multiple pathways such as substrate degradation, energy metabolism, reproductive growth and response to environmental changes.

### 3.4. Differential Expression of Carbohydrate Enzyme Genes in Different Growth Developmental Stages

Carbohydrate-active enzymes are a large class of important enzymes divided into six types, namely, glycoside hydrolases (GHs), carbohydrate esterases (CEs), carbohydrate-binding modules (CBMs), auxiliary activities (AAs), glycosyl transferases (GTs) and polysaccharide lyases (PLs), and they have functions such as degradation, modification and glycosidic bond formation [29]. Therefore, we tracked the expression of carbohydrate enzyme genes during the process of mycelial vegetative growth to the fruiting body stage (Figure 4). GHs play an important role in the hydrolysis and synthesis of sugars and glycoconjugates in organisms. In the GH family, the expression patterns during growth and development were divided into three types: 12 GH family genes were significantly upregulated in the G-S stage, 5 genes were significantly upregulated in the P-S stage and 4 genes were significantly upregulated in the M-F stage (Figure 4A). In the CE family, nine genes were significantly upregulated in the G-S stage, three genes were significantly upregulated in the P-S stage and two genes were significantly upregulated in the M-F stage (Figure 4B). CBMs mainly improve the catalytic efficiency of carbohydrate active enzymes. Among these genes, nine had the highest expression in the G-S stage, two had the highest expression in the P-S stage and CBM19 (DQGG009537) had the highest expression levels in the M-F stage (Figure 4C). AA family enzymes are also key enzymes that improve degradation efficiency. Ten genes were significantly upregulated in the G-S stage, and the AA7 (DQGG005602), AA9 (DQGG010609) and AA5 (DQGG007622) genes were also significantly upregulated in the M-F stage (Figure 4D). In addition, the GT and PL genes were much less abundant than other carbohydrate enzymes. In the GT family, seven genes were significantly upregulated in the M-F stage, while the GT48 (DQGG007250), GT2 (DQGG006597) and GT2 (DQGG006533) family genes were upregulated in the G-S stage (Figure 4E). Moreover, five genes were significantly upregulated in the G-S stage, and PL4 (DQGG005928) and PL14 (DQGG010936) were also significantly upregulated in the P-S stage (Figure 4F). This result indicates that a large number of carbohydrate enzyme (i.e., GH, CE, CBM, AA and PL) genes are upregulated during the vegetative growth stage of mycelium and mainly play a role in substrate degradation. However, a small number of genes were more highly expressed in the fruiting body development stage, indicating that they might also participate in fruiting body development.

### 3.5. DEGs Related to Primordium Formation during Growth Developmental Stages

The CO_2_ concentration and temperature in the growth environment of edible fungi are the key conditions affecting the growth of mycelium and the formation and development of fruiting bodies. Therefore, specific analyses of CO_2_-responsive genes, temperature-responsive genes and transcription factors regulating fruiting body development were performed. Carbonic anhydrase (CA, EC4.2.1.1) is a type of zinc metalloenzyme that can efficiently catalyze the reversible reaction between CO_2_ and water to produce carbon acid and hydrogen protons that regulate the balance of CO_2_/HCO_3_^−^ in the cell [30]. Therefore, CA is an important enzyme enabling the mycelium to sense CO_2_. A total of three CA genes were obtained in the genome of *S. rugosoannulata. CA-1* (DQGG004716) had the highest expression level in the G-S stage, while its expression level was downregulated in other stages, and *CA-2* (DQGG004326) and *CA-3* (DQGG008753) also had the highest expression levels in the M-F and P-S stages, respectively (Figure 5B). Studies have shown that CO_2_ can regulate the growth of mycelium through the cAMP signaling pathway [31]. Accordingly, we found that two adenylyl cyclase genes (i.e., DQGG003026 and DQGG004524) that regulate cAMP synthesis were upregulated at the P-S and M-F stages (Figure 5C). However, the cAMP-dependent protein kinase (PKA) genes downstream of cAMP regulation were further found to be differentially expressed, of which two genes (i.e., DQGG007390 and DQGG001349) were upregulated at the P-S and M-F stages, and the other two (i.e., DQGG000595 and DQGG004694) were downregulated (Figure 5D).

Heat shock proteins (HSPs) are typical proteins that respond to temperature changes to prevent damage to the mycelium caused by high temperature. The hsp90-domain-containing protein (DQGG001016), hsp70 (DQGG010653), hsp88 (DQGG004214) and hsp31 (DQGG001388) were the most highly expressed in the G-S stage among all HSP DEGs. Low temperature (12 °C) is an effective stimulus for the formation of primordia, and we found that most genes were significantly downregulated in the P-S stage (Figure 6A). However, 11 more HSP genes were upregulated in the M-F stage, suggesting that HSPs may be involved in resistance to environmental stress during growth and development.

Transcription factors are key proteins that ensure that the target gene is expressed at a specific time and location with a specific intensity. In the P-S stage, 11 transcription factors had higher expression levels. At the same time, there were six transcription factors with the highest expression in the M-F stage (Figure 6B). Notably, the transcription factors that might be involved in the regulation of growth and development mainly included steA (i.e., DQGG000640 and DQGG001965), MYB (i.e., DQGG003605), C6 finger domain-containing nosA (i.e., DQGG002798), HAP1 (i.e., DQGG002147) and GATA-4/5/6 (i.e., DQGG008309).

### 3.6. Validation of Transcriptomics Data by RT–qPCR

Based on gene expression patterns, typical genes involved in growth and development and detected as DEGs were selected for qRT–PCR analysis. Among the CAZymes, nine enzyme genes were identified: GH7 (i.e., DQGG002368), GH11 (i.e., DQGG004511), CE1 (i.e., DQGG010182), CE5 (i.e., DQGG000985), CBM5 (i.e., DQGG006344), AA9-1 (i.e., DQGG009829), AA9-2 (i.e., DQGG009624), GT2 (i.e., DQGG006597) and PL14 (i.e., DQGG010936) (Figure 7A). Most of these enzyme genes were expressed at the highest level in the G-S stage, similar to the transcriptome expression pattern. In addition, we also verified the expression patterns of two carbonic anhydrases (i.e., *CA-1* (DQGG004716) and *CA-2* (DQGG008753)), two cAMP-dependent protein kinases (i.e., *PKA-1*(DQGG004293) and *PKA-2* (DQGG007322)), two HSP genes (i.e., *hsp60* (DQGG011020) and *hsp90* (DQGG001016)) and two transcription factors (*TF-MYB* (DQGG003605) and *TF-STEA* (DQGG001965)) (Figure 7B–E). This result indicates that these qRT–PCR expression patterns and the RNA-Seq results at different developmental stages have similar trends, further validating the RNA-Seq results.

## 4. Discussion

*S. rugosoannulata* is widely cultivated in China because of its easy cultivation and extensive management and the use of agricultural waste as a substrate for growth [32]. *S. rugosoannulata* not only has high commercial value but also has a certain medicinal value (i.e., preventative effect against coronary heart disease, hyperglycemia and solid tumor S-180) [33]. However, because *S. rugosoannulata* can be cultivated only according to seasonal temperature changes, such as in Eastern China, the mushroom is generally cultivated from November to December in the first year and harvested from March to April in the second year [7]. Therefore, the inability to industrialize production has always been an important factor restricting the development of *S. rugosoannulata.* To improve industrial production, we explored a factory cultivation model and used developmental transcriptomics to study the growth and development of *S. rugosoannulata.*

In the present study, the transcriptomes of different developmental stages of *S. rugosoannulata* were compared using the same batch of samples. We compared vegetative growth at the hyphal (G-S) and primordium (P-S) stages and found 2696 DEGs, of which 1769 were upregulated and 1290 were downregulated. From the primordium stage (P-S) to the mature fruiting body stage (M-F), a total of 3287 DEGs were identified, and 50.5% were upregulated. Previous studies have found that *C.*
*cordyceps*, *C. cinerea* and *Pleurotus eryngii* have a large number of DEGs when hyphal and primordia transcriptomes are compared, which is similar to our research results [22,26,34]. However, we found more DEGs in the comparison between the primordium and the mature fruiting body, while the number of DEGs found in other species was lower. These results suggest that a large number of DEGs are involved in the formation of complex phenotypes of fruiting bodies. Transcriptome DEG enrichment analysis is usually used in fruiting body growth and development studies, such as in *C.*
*cordyceps* [26], *Morchella*
*importuna* [35] and *C.*
*cinerea* [22]. The GO enrichment analysis showed that hydrolase activity, small molecule metabolic process, small molecule biosynthetic process and carbohydrate-derivative metabolic process were significantly enriched in the comparison between the G-S and P-S stages. Moreover, hydrolase activity was also significantly enriched in the hyphal stage of *C.*
*cordyceps* [26]. Thus, substrate degradation by the vegetative mycelium requires more hydrolytic enzymes. However, more genes involved in structural molecule activity, organelle fission, ribosome, mitochondrial protein complex, structural constituent of ribosome and meiosis I were enriched during the P-S to M-F stages. This result indicates that in the development from the primordium to the fruiting body, continuous cell proliferation and differentiation are required to promote the growth and development of the fruiting body and gradually carry out the process of sexual reproduction.

Carbohydrate-active enzymes degrade specific substrates (lignocellulose) and are also involved in the growth and development of mushrooms and the synthesis of plant lignocellulose [36,37]. In our research, we found that most carbohydrate enzymes, such as exoglucanase (GH7), endo-1,4-β-xylanase (GH11), acetylxylan esterase (CE5), carbohydrate-binding module family 5 protein (CBM5), glycoside hydrolase family 61 protein (AA9), glycosyltransferase family 2 protein (GT2) and polysaccharide lyase family 14 protein (PL14), were upregulated during the mycelial growth stage. In a study of *A. bisporus*, compost-grown mycelium expressed a large diversity of CAZyme genes related to the degradation of plant biomass components [5,36]. This finding is similar to the process of *S. rugosoannulata* degrading the straw substrate during the mycelial growth stage. However, we also found that some CAZyme genes, such as GH5 (DQGG007848), GH47 (DQGG009025), CE1 (DQGG003582), CBM13 (DQGG011341), AA9 (DQGG010609), GT2 (DQGG004795), GT48 (DQGG002416) and PL14 (DQGG010936), had higher expression levels in the primordium stage and mature fruiting body stage. Among these genes, GT2 (DQGG004795) and GT48 (DQGG002416) belonged to chitin synthase and 1,3-β-glucan synthase, respectively. These two enzymes are also key enzymes in the synthesis of chitin and glucan in fungal cell walls [37], also implying that CAZyme genes may be involved in cell wall formation. Studies of *A. bisporus* have also found that multiple sugar degradation pathways are involved in the mycelial stage, but only glycolysis pathways are involved in the fruiting body stage, and the transfer of accumulated sugars to the fruiting body may involve specific carriers and transporters [5,36]. We also found that CAZyme genes are involved in the sugar transport process in the fruiting body stage of *S. rugosoannulata.* Therefore, the differential expression of carbohydrate and enzyme genes is not only involved in the degradation of substrates but may also be involved in nutrient utilization and cell wall synthesis.

Environmental factors that influence fruiting body induction in basidiomycetes individually or in combination include physical (i.e., light, temperature and injury) and physiological (i.e., nutrients, gaseous components and hormones) factors [8]. However, we further studied the fruiting body development of *S. rugosoannulata* under the conditions of stable nutrition and strict control of CO_2_ concentration, temperature, humidity and light. Our research revealed that the genes related to fungal CO_2_ sensing were differentially expressed. In edible fungi, there are few reports on the response to CO_2_ changes and the regulation of downstream gene expression. To date, seven carbonic anhydrase superfamily protein genes related to CO_2_ regulation have been reported in *F. velutipes* [30]. In our study, only three genes encoding carbonic anhydrase protein were found: *CA1* (DQGG004716) had the highest expression levels in the vegetative hyphae, and the highest expression levels of *CA2* and *CA3* were observed in the primordia and fruiting bodies stages, respectively, which may be the result of the response to CO_2_ through multiple genes. In addition, studies have shown that CO_2_ can activate adenylyl cyclase, leading to increased cAMP levels. Subsequently, the cAMP pathway was activated, which controls a variety of cellular processes including stress response and metabolism [31]. Another study found that the cAMP pathway was also involved in the growth and development of *Hypsizygus marmoreus* [38]. Interestingly, the expression of the adenylyl cyclase gene was also upregulated during the formation of primordia and fruiting body development of *S. rugosoannulata*, and the PKA gene regulated by cAMP was also significantly upregulated. However, measuring the cAMP content of the straw mushroom growth and development stage also showed that the highest content occurred in the primordium stage [39], therefore suggesting that the accumulation of cAMP caused by a series of enzymatic reactions induced by CO_2_ may play an important role in primordium formation and fruiting body development.

Temperature change is also a key factor stimulating primordium formation [8]. This study showed that the HSP genes were differentially expressed during the growth and development of *S. rugosoannulata.* The expression of most genes was significantly downregulated during the formation of the primordium stage by a temperature downshift (22–12 °C). At the same time, the expression of HSP genes were significantly upregulated in the mature fruit body stage (Figure 6A). A temperature decrease is one of the necessary conditions for the formation of fruiting bodies of *L. edodes* [13], *A. bisporus* [14], *F. velutipes* [15] and *P. eryngii* subsp. *tuoliensis* (Bailinggu) [40]. The HSPs in *S. rugosoannulata* were significantly affected by temperature, and mycelium resistance to low-temperature stress may also be a key condition for stimulating the formation of primordia. We also searched for cold shock proteins, and a previous study showed that *F. velutipes* presents FDS protein genes that respond to temperature decreases [15]; however, none of these genes were found among the annotated genes from transcriptome data. Therefore, this aspect requires further study.

Transcription factors are involved primarily in the regulation of cell meiosis, growth and development, primary metabolism, secondary metabolism, and drug resistance in fungi [41,42]. Therefore, they are expected to be very important for regulating the formation, growth and development of *S. rugosoannulata.* We found that in the entire life cycle of *S. rugosoannulata*, most of the transcription factors were expressed at a low level at the hyphal stage but at the highest level at the primordium stage, such as zinc finger transcription factor (DQGG001624), transcription factor steA (DQGG000640 and DQGG001965), transcription factor MYB (DQGG003605), C6 finger domain transcription factor nosA (DQGG002798), transcription factor HAP1(DQGG002147) and GATA-4/5/6 transcription factor (DQGG008309). In rice blast fungus, Zn_2_Cys_6_ transcription factor genes involved in pathogenicity frequently tend to function in multiple developmental stages [43]. In plants, TF-MYB has been found to regulate plant growth and differentiation [44]. In *Aspergillus*, the transcription factor steA is essential for sexual reproduction, while the transcription factor nosA also plays an important role in stress responses and development [45,46]. In addition, the transcription factor HAP1 regulates the expression of oxygen-dependent genes, which have a crucial role in growth, and GATA-4/5/6 transcription factors are also able to regulate animal cell differentiation [47,48]. Previous research has shown that transcription factors affect mycelial growth and both asexual and sexual development in *Neurospora* [49]. Therefore, transcription factors (i.e., steA, MYB, nosA, HAP1 and GATA-4/5/6) may have important roles in the formation, development and sexual reproduction of fruiting bodies of *S. rugosoannulata*.

## 5. Conclusions

In conclusion, our results revealed the differential expression patterns of genes in the mycelium, primordium and fruiting body stages of *S. rugosoannulata* under factory cultivation conditions. Moreover, we detected 9690 DEGs in different growth and development stages. Furthermore, we also found that CAZyme genes not only degrade the substrate at the hyphal stage but may also play a role in the formation of cell walls during development. Carbonic anhydrase responds to changes in CO_2_ concentration, and combined with mediation of the synthesis of cAMP and HSPs in response to temperature changes, might also play a key role in primordium formation and fruiting body development. In addition, we found that transcription factors are necessary for the regulation of primordium formation. Overall, this report presents the first detailed developmental transcriptomic study of *S. rugosoannulata*. These results will improve our understanding of the nutritional and environmental factors that promote fruiting body formation and provide a foundation for improving the industrialized cultivation of *S. rugosoannulata*.

## Figures and Tables

**Figure 1 genes-13-01080-f001:**
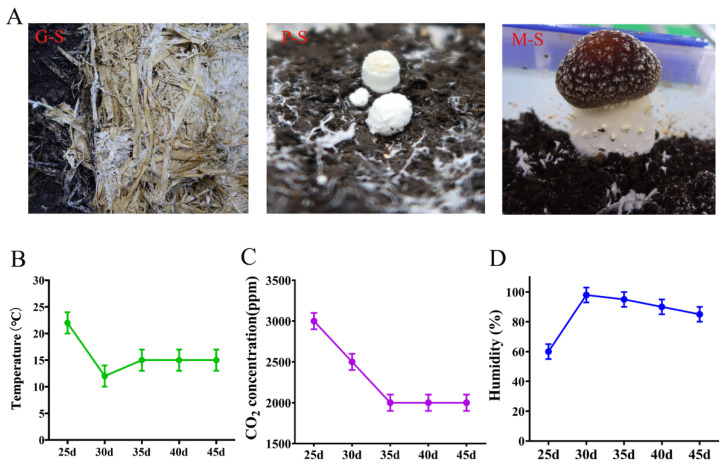
Different growth and development stages of *S. rugosoannulata* under industrialized conditions. (**A**) Different developmental stages of *S. rugosoannulata*—G-S: hyphal growth stage; P-S: primordium stage; M-F: mature fruiting body stage. (**B**) Temperature changes during growth and development. (**C**) Changes in carbon dioxide concentration during growth and development. (**D**) Humidity changes during growth and development.

**Figure 2 genes-13-01080-f002:**
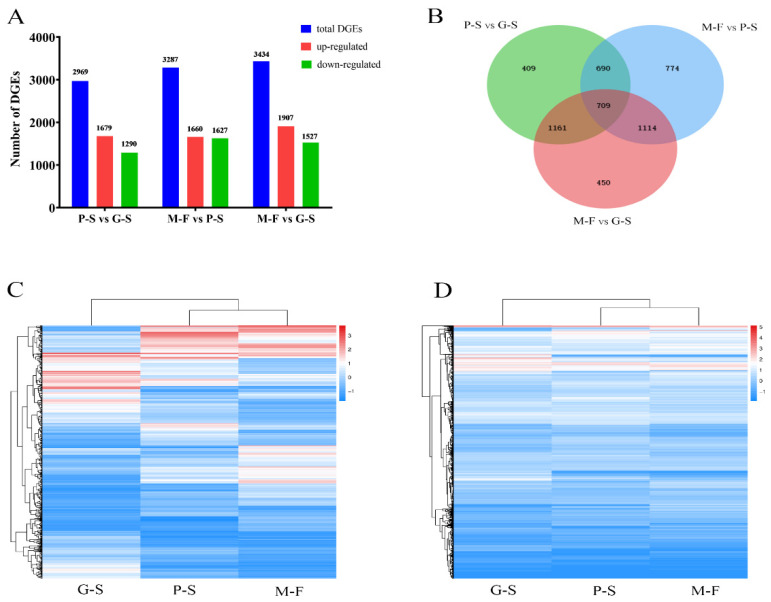
Analysis of DEGs at different growth and development stages. (**A**) DEG distribution between the two samples analyzed. The number of DEGs is indicated at the top of the histograms. (**B**) Venn diagrams comparing shared DEGs among the different growth stages. (**C**) Expression patterns of the 709 genes that overlap between P-S vs. G-S, M-F vs. P-S and M-F vs. G-S. (**D**) Gene expression levels in different growth and development stages.

**Figure 3 genes-13-01080-f003:**
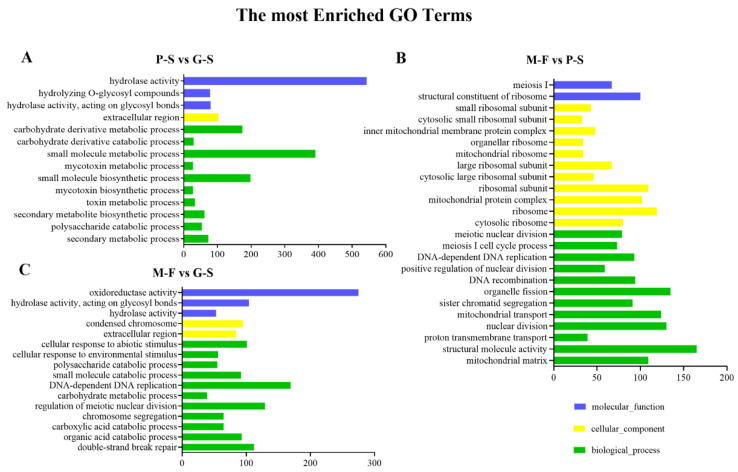
GO functional classification of DEGs. (**A**–**C**) GO enrichment between different growth and development stages. The green bars represent biological processes; yellow bars represent cellular components; blue bars represent molecular functions. Only the significant GO terms (*p* < 0.005) are shown.

**Figure 4 genes-13-01080-f004:**
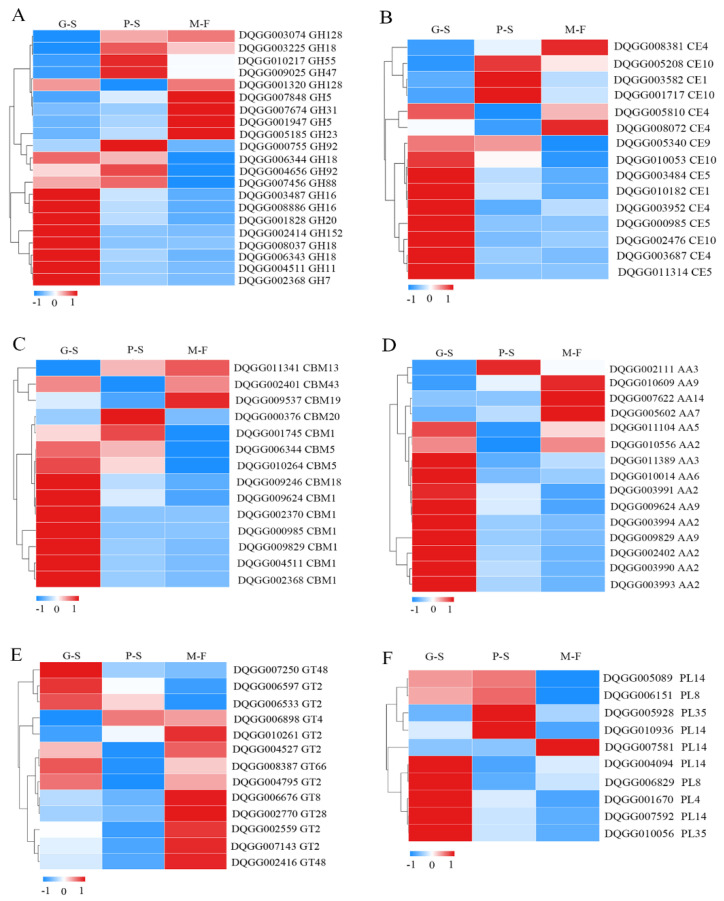
Differential expression of carbohydrate enzyme genes in different developmental stages: (**A**) differential expression of glycoside hydrolase (GH) genes; (**B**) differential expression of carbohydrate esterase (CE) genes; (**C**) differential expression of carbohydrate-binding module (CBM) genes; (**D**) differential expression of auxiliary activity (AA) genes; (**E**) differential expression of glycosyl transferase (GT) genes; (**F**) differential expression of polysaccharide lyase (PL) genes.

**Figure 5 genes-13-01080-f005:**
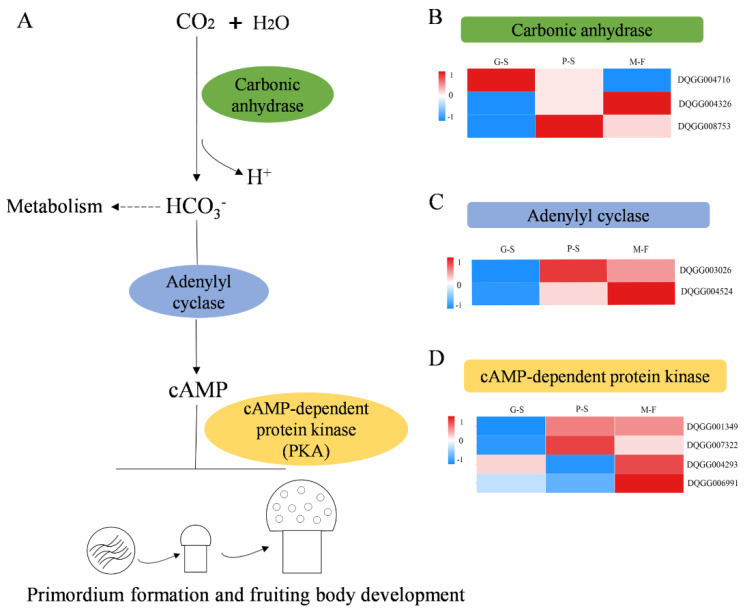
Expression patterns of carbon dioxide sensing-related genes: (**A**) schematic diagram of the involvement of carbon dioxide in the formation of primordia and development of fruiting bodies; (**B**) expression patterns of carbonic anhydrase genes; (**C**) expression pattern of adenylyl cyclase genes; (**D**) expression pattern of cAMP-dependent protein kinase (PKA) genes. The levels of expression are represented by log_2_(FPKM + 1) values after centralization correction.

**Figure 6 genes-13-01080-f006:**
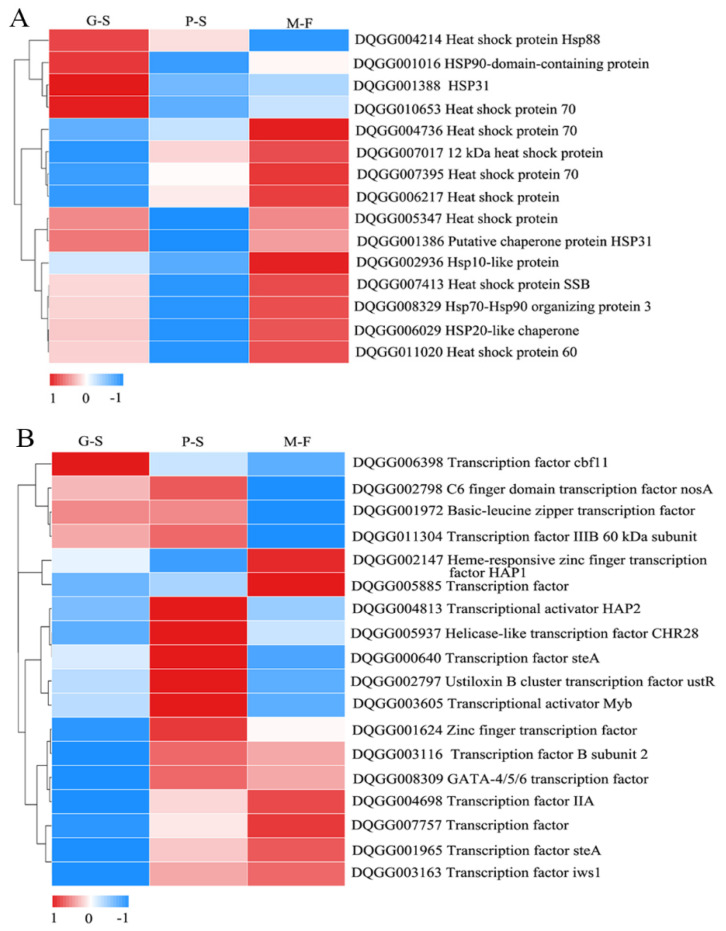
Heatmap analysis of the expression patterns of heat shock proteins and transcription factors: (**A**) differential expression of heat shock proteins in response to temperature changes; (**B**) differential expression of transcription factors during growth and development. The expression levels are represented by log_2_(FPKM + 1) values after centralization correction. Genes with similar patterns of expression are clustered together.

**Figure 7 genes-13-01080-f007:**
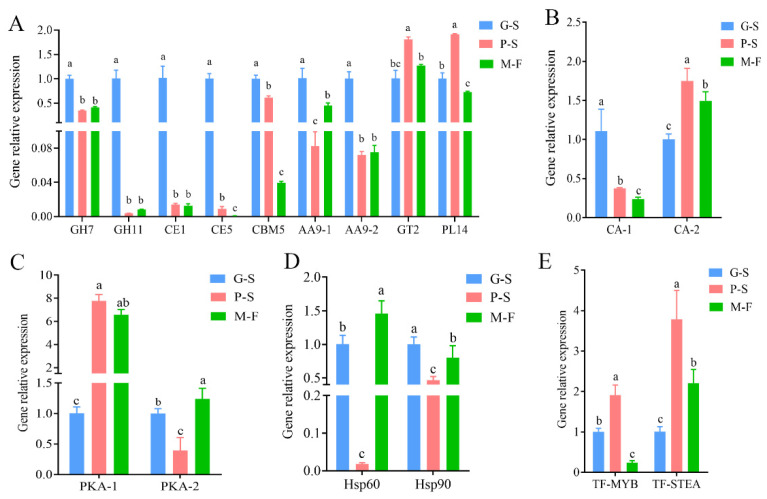
Validation of the gene expression levels of specific enzymes: (**A**) ten differentially expressed CAZyme genes (i.e., *GH7*, *GH11*, *CE1*, *CE5*, *CBM5*, *AA9-1*, *AA9-2*, *GT2* and *PL14*) in the growth and development stages; (**B**) two differentially expressed carbonic anhydrase genes (i.e., *CA-1* and *CA-2*) in the growth and development stages; (**C**) two differentially expressed PKA protein kinase genes (i.e., *PKA-1* and *PKA-2*) in the growth and development stages; (**D**) two differentially expressed heat shock protein genes (i.e., *hsp60* and *hsp90*) in the growth and development stages; (**E**) two differentially expressed transcription factor genes (i.e., *TF-MYB* and *TF-STEA*) in the growth and development stages. All data are presented as the means ± standard deviations (SDs) of three independent experiments. Bars with different letters are significantly different at *p* < 0.05 according to Duncan’s multiple range test.

## Data Availability

Not applicable.

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
