# Peer review of "Transcriptome and Differentially Expressed Gene Profiles in Mycelium, Primordium and Fruiting Body Development in Stropharia rugosoannulata"

_genes, 2022, doi:10.3390/genes13061080_

Round 1

Reviewer 1 Report

Please have a native English speaker or an experienced researcher to help you with the writing. From the annotated manuscript, those highlight spots showed either bad English writing or wrong scientific expression. The authors should revisit some basic knowledges of meiosis and meiosis specific gene and expression, in mushroom-forming fungi. Mushrooms have long dikaryotic stage but short diploid stage followed by meiosis that happen only in basida and basidiospore. What is your point to study this process in the vegetative growth and pre-sporulation? BTW, transcriptomics data from this study is of some interest to the society.

Author Response

Reviewer #1: Please have a native English speaker or an experienced researcher to help you with the writing. From the annotated manuscript, those highlight spots showed either bad English writing or wrong scientific expression. The authors should revisit some basic knowledges of meiosis and meiosis specific gene and expression, in mushroom-forming fungi. Mushrooms have long dikaryotic stage but short diploid stage followed by meiosis that happen only in basidia and basidiospore. What is your point to study this process in the vegetative growth and pre-sporulation? BTW, transcriptomics data from this study is of some interest to the society.

Response: We are very grateful for your questions to make the manuscript clearer.

(1) We have carefully modified the highlighted spots and sent the revised manuscript to edit the language by American Journal Experts and the Certificate Verification Key is F035-DCF0-B7A8-7A91-FD5E.

(2) Thank you again for reminding us to revisit the knowledge regarding the meiosis process. Because the meiosis pathway was significantly enriched in the M-F vs. P-S comparison, we were concerned that this pathway was involved in the sexual reproduction process of S. rugosoannulata. Differential expression analysis of meiosis pathway genes at the vegetative growth stage (G-S) was not the focus and was treated as a control result. We focused on the differentially expressed genes of the meiosis pathway at the mature fruiting body stage (M-F), which may be involved in spore formation (sexual reproduction).  

We also understand that meiosis occurs during the formation of basidia and basidiospores. However, the objective of this study was to identify the key pathways and genes that affect primordia formation, fruiting body development and sexual reproduction through transcriptome analysis of the three growth stages of S. rugosoannulata. Moreover, some meiosis pathway genes were significantly upregulated during fruiting body maturation (M-F). This suggests that they may be involved in basidiospore formation and fruiting body development. This accidental discovery only provides some insight into the sexual reproduction and fruiting body development of S. rugosoannulata, and the influence of meiosis pathway genes on sporogenesis needs further study. We also further discuss the phenomenon of upregulated expression of many meiosis pathway genes at the M-F stage (lines 468-471).

Reviewer 2 Report

The authors performed RNA-seq analysis of the three stages: the mycelium, primordium, and fruiting body, to identify DEGs in Stropharia rugosoannulata.

After minor revisions, the paper would be better.

Minor points

Lines 194-195: The carbon dioxide concentration was measured. The authors should mention how to measure the carbon dioxide concentration in the Materials and Methods section.

Line 158: The sentence should be rewritten.

Author Response

Reviewer #2: The authors performed RNA-seq analysis of the three stages: the mycelium, primordium, and fruiting body, to identify DEGs in Stropharia rugosoannulata. After minor revisions, the paper would be better.

Minor points

Lines 194-195: The carbon dioxide concentration was measured. The authors should mention how to measure the carbon dioxide concentration in the Materials and Methods section.

Response 1: We appreciate these suggestions. We have added a detailed description of the temperature, humidity and carbon dioxide concentration records to the Materials and Methods (lines 116-119).

Line 158: The sentence should be rewritten.

Response 2: We appreciate this suggestion. This sentence has been rewritten (lines 148-149).

Round 2

Reviewer 1 Report

I read the revised manuscript and the authors' responses to my comments. While the manuscript was improved in writing, the content in this revision still is problematic. The major point is that the authors claimed wrong points about their discoveries. Meiosis process has been intensively studied in fungal models, and it is well known and fully expected that core meiosis genes are expressed during sexual development in sexual spore formation. To claim that meiosis was actively up-regulated during sexual development is bascially nonsense. I pointed out that meiosis is a very quick process during mushroom spore development, and the data from this study basically can not provide high quality data for studying genetics meiosis. The study only provided data of minimum interest on "HOW" meiosis genes and pathways may be regulated during sexual development in this specific mushroom-forming fungus. Moreover, meiosis genes, except the core meiosis genes, are shared with the mitosis process, and it is not surprising at all that those genes are also expressed during other growth and developmental stages. The authors should completely remove the meiosis part of this study, or to collect more data and to run more specific analysis regarding the meiosis in this fungus.

Talking about the writing, the revision seems much better, but there are still some issues. The title is quite confusing, and it probably is better and more accurate as "Transcriptome and differentially expressed genes profiles in mycelium, primordium and fruiting body development in Stropharia rugosoanulata". Lines 33-34 were meaningless to compare qRT-PCR with RNAseq results. Line 60, it should be "xxx is similar to xxx, and both fungi use straw...", Line 70 "conserved" is not clear. What did you mean by "conserved"? regarding genes gains and losses? or gene sequences and functions? Lines 86-87, please provide reference(s). Line 91-92, should be "However, unlike in the ascomycetes, the molecular mechanism for the fruiting body development in basidiomycetes is less well understood"... etc.

Author Response

1、I read the revised manuscript and the authors' responses to my comments. While the manuscript was improved in writing, the content in thi revision still is problematic. The major point is that the authors claimed wrong points about their discoveries. Meiosis process has been intensively studied in fungal models, and it is well known and fully expected that core meiosis genes are expressed during sexual development in sexual spore formation. To claim that meiosis was actively up-regulated during sexual development is bascially nonsense. I pointed out that meiosis is a very quick process during mushroom spore development, and the data from this study basically can not provide high quality data for studying genetics meiosis. The study only provided data of minimum interest on "HOW" meiosis genes and pathways may be regulated during sexual development in this specific mushroom-forming fungus. Moreover, meiosis genes, except the core meiosis genes, are shared with the mitosis process, and it is not surprising at all that those genes are also expressed during other growth and developmental stages. The authors should completely remove the meiosis part of this study, or to collect more data and to run more specific analysis regarding the meiosis in this fungus.

Response: We appreciate these suggestions. We also agree with you that meiosis is a very quick process during mushroom spore development. These data cannot explain this process well. Therefore, we have removed the meiosis part of this study.

2、(1) Talking about the writing, the revision seems much better, but there are still some issues. The title is quite confusing, and it probably is better and more accurate as "Transcriptome and differentially expressed genes profiles in mycelium, primordium and fruiting body development in Stropharia rugosoanulata".

Response: We appreciate these suggestions. We have carefully modified the manuscript and sent the revised manuscript to edit the language by American Journal Experts and the Certificate Verification Key is F035-DCF0-B7A8-7A91-FD5E. In addition, we have modified the title of the manuscript according to your suggestion (Lines 1-3).

(2) Lines 33-34 were meaningless to compare qRT-PCR with RNAseq results.

Response: We appreciate these suggestions. We have deleted this sentence.

(3) Line 60, it should be "xxx is similar to xxx, and both fungi use straw..."

Response: We appreciate this suggestion. We have rewritten this sentence (Lines 56-57).

(4) Line 70 "conserved" is not clear. What did you mean by "conserved"? regarding genes gains and losses? or gene sequences and functions?

Response: We are very grateful for your questions that help make the manuscript clearer. "Conserved" refers to whether the composition and number of CAZyme genes are similar with those of other litter/straw-degrading species (A. bisporus, Volvariella volvacea and Coprinopsis cinerea). We have rewritten this sentence (Lines 66-68).

(5) Lines 86-87, please provide reference(s).

Response: We appreciate this suggestion. We have added references (Line 85).

(6) Line 91-92, should be "However, unlike in the ascomycetes, the molecular mechanism for the fruiting body development in basidiomycetes is less well understood"... etc.

Response: We appreciate this suggestion. We have rewritten this sentence (Lines 89-90).